# Malignant Epithelioid Neoplasm versus Dedifferentiated Malignant Melanoma: A Case Report

**DOI:** 10.3390/diseases12090196

**Published:** 2024-08-24

**Authors:** Angela Rosenberg, Chapman Wei, Yisroel Grabie, Stephanie Chain, Sakura Thapa, Gita Vatandoust

**Affiliations:** 1Department of Medicine, Northwell Health—Staten Island University Hospital, Staten Island, NY 10305, USA; arosenberg10@northwell.edu (A.R.);; 2Department of Pathology, Northwell Health—Staten Island University Hospital, Staten Island, NY 10305, USA

**Keywords:** epithelioid malignant neoplasm, poorly differentiated epithelioid carcinoma, metastatic tumor, seborrheic keratosis, malignant melanoma

## Abstract

The metastasis of poorly differentiated epithelioid carcinoma to the axillary node is uncommon. This tumor has heterogeneous expression and is challenging to diagnose with certainty. Often, it necessitates immunoperoxidase staining to ascertain the tumor lineage, and diagnosis is prolonged due to low suspicion. Herein, we present a case involving a 75-year-old male war veteran with a prior history of a gunshot wound complicated by colostomy that presented with an axillary mass, fecal and urinary incontinence, leg weakness, fevers, night sweats, and substantial weight loss. On admission, he had heightened leukocytosis (43K), anemia (hemoglobin 6.6), and thrombophilia (1000). This patient constantly picked at his back to remove recurrent “gun shrapnel” eruptions. An excisional biopsy of the axillary mass was performed for diagnosis and lymph node removal. Notably, after excision, there was marked improvement in the presenting symptoms. Diagnostic challenges arose due to the tumor cells’ inconsistent immunohistochemical marker expression. The staining patterns alluded to metastatic melanoma. Yet, the tumor displayed epithelial characteristics, supported by an immunophenotypic marker pattern indicative of poorly differentiated carcinoma. This case underscores the morphological and immunoperoxidase staining similarities between poorly differentiated carcinoma and dedifferentiated tumors of varying origins. It illustrates the intricate nature of these malignant metastatic tumors and their overlapping manifestations, which requires provider awareness. The timely diagnosis of poorly differentiated epithelial carcinoma remains paramount to early treatment and improved prognosis. Therefore, in patients manifesting with an axillary mass, fecal and urinary incontinence, and B-symptoms, poorly differentiated epithelial carcinoma should be included in the differential diagnosis.

## 1. Introduction

Metastatic epithelial carcinomas exhibit a diverse range of antigen expressions due to their poorly differentiated nature [1]. A poorly differentiated malignant tumor can signify either an enigmatic metastasis from an unidentified source or primary neoplasia lacking a clear cell lineage of differentiation. Due to the wide array of potential entities encompassed within each category, including those from epithelial, mesenchymal, hematopoietic, or melanocytic lineages in the differential diagnosis, the diagnostic process can be time-consuming and may result in hospital course delays [2]. An accurate diagnosis requires specialized pathologic evaluation with immunoperoxidase staining and electron microscopy [3]. Since complex histochemical signatures arise from malignant dedifferentiated tumors, biomarkers or genomic analysis may be required to confirm the diagnosis of poorly differentiated carcinoma [4].

We describe a case of a 75-year-old male with fecal and urinary incontinence, unintentional weight loss, weakness, persistent fevers, night sweats, and a right axillary mass. Staining of the initial incision and drainage strongly indicated metastatic melanoma. However, the excisional biopsy displayed morphology and immunophenotypic characteristics in line with poorly differentiated epithelioid carcinoma. This diagnostic discrepancy led to delays in commencing treatment, underscoring the need for a comprehensive review, given the overlapping features between poorly differentiated carcinoma and dedifferentiated melanoma.

## 2. Case Report

A 75-year-old male, a Vietnam War veteran, sought medical attention at Staten Island University Hospital in New York City due to a persistent one-month history of unrelenting fevers, night sweats, substantial weight loss, and the emergence of a rapidly growing, painful mass in his right axilla. Notably, his medical history included a gunshot wound that had necessitated a colostomy, a procedure that had been subsequently reversed. He had not sought medical care since the colostomy reversal, spanning over five decades.

Upon arriving at the emergency department from urgent care, the patient displayed tachycardia (heart rate of 112), leukocytosis (43,000), thrombocythemia (841,000), anemia (hemoglobin 6.6), thrombophilia (1000), transaminitis, and an elevated troponin level (0.4). These findings may be seen in myeloproliferative neoplasms (thrombocythemia, myelofibrosis, and polycythemia vera) which cause the overproduction of blood cells. A computed tomography scan showed a right axillary soft tissue mass measuring 8.3 × 3.9 × 6.5 cm with multiple prominent retroperitoneal lymph nodes, suggestive of a neoplastic process (Figure 1).

An incision and drainage of the axillary mass had been performed on day two of hospitalization by the surgery department. The specimen was sent to our affiliated pathology department for staining and microscopic analysis. The neoplastic cells seen in the specimen were positive for WT1, SOX10, and CAM5.2 with CD45 positive on a few lymphocytes. The cells also tested negative for AE1/AE3, CK7, CK20, TTF-1, Melan-A, HMB45, CDX2, CD138, chromogranin, synaptophysin, S100, mammaglobin, GATA3, and GCDFP (Table 1). Due to its conflicting immunophenotype, the sample was sent to an outside institution for collaboration.

Staining from an outside institution showed NRASQ61R, BerEp4, CK7, CD30, GFAP, GATA3, BRAFv600E, and p63m EMA were negative. INI-1 and SMARCA4 were positive (i.e., normal/retained) (Table 2). Although it is very non-specific, PRAME was found to have diffuse nuclear positivity.

Microscopically, the specimen was a cellular, cytologically malignant spindle cell neoplasm composed predominantly of epithelioid cells with large atypical vesicular nuclei and prominent nucleoli with somewhat amphophilic cytoplasm (Figure 2). There was multifocal strong positivity for SOX10 (Figure 3), pan-keratin, and CAM5.2 (Figure 4), along with negativity for S100, HMB45, and Mart-1. Multifocal positivity was also found in claudin-4 with a distinctively membranous pattern, strongly suggesting epithelial differentiation. The specimen was classified as a poorly differentiated epithelioid malignant neoplasm, suggestive of metastatic carcinoma.

On the 12th day of his hospitalization, the patient experienced new onset bowel incontinence. Following an excisional biopsy of the axillary mass on the 28th day of admission, his white blood cell count notably halved to 20,000, and his bowel incontinence improved. The excised skin fragment exhibited dimensions of 8 × 6 cm. Within the tissue, a focal region of reddish-brown discoloration spanning 5 × 5.5 × 3 cm was observed, while the rest of the tissue displayed a tan-white coloration and a lobulated structure. The underlying fibroadipose tissue measured 11 × 8 × 3 cm, resulting in a total excisional mass weight of 283 g. The repeat biopsy was sent to Foundation Medicine, Inc., but there were no reportable alterations with companion diagnostic claims.

Biomarker findings (Table 3) included a microsatellite status of MS-stable and a tumor mutational burden of 4 muts/mb. Genomic findings were MAP2K1 (MEK1) E102_I103del, MYC amplification, CDKN2A/B CDKN2B loss, p16INK4a loss, and p14ARF loss in exons 2-3, PIK3C2G E1231K, RB1 Q395*, TERT promoter -146C>T, and TET2 D1788fs*32. Variants of unknown significance were also located in the specimen, but they might not have been adequately characterized in the scientific literature at the time the pathology report was issued and/or the genomic context of these alterations also makes their significance unclear. These variants included ATR, V1689M, ATRX amplification, FAM123B, R21H, KIT, T67S, PDGFRB rearrangement, ZNF703, and A401_H402insPTHLGGSSCST CSA. 

The patient was diagnosed with poorly differentiated epithelioid carcinoma of the axilla. The retroperitoneal lymph nodes were never biopsied, as the multifocal involvement pointed less to a primary neoplasm and would likely not have changed management. After a discussion of diagnosis, prognosis, and goals of care, the patient preferred no treatment.

## 3. Discussion

The presented case falls under the umbrella terms of poorly differentiated malignant neoplasm, which usually arises as a metastasis of unknown origin primary neoplasia lacking a clear cell lineage of differentiation. The exact incidence of poorly differentiated carcinoma (PDC) originating from an unknown primary tumor site remains elusive, and there is also uncertainty regarding the proportion of highly treatable tumors within this category [5]. Remarkably, the presence of axillary lymph node metastases originating from adenocarcinoma or poorly differentiated carcinoma of unknown primary (CUPAx) emerged as a rarity within the clinical landscape. The scarcity of consensus surrounding its underlying biology, optimal management strategies, and prognosis underscores the complexity and enigmatic nature of this clinical entity.

Metastatic carcinomas spread to distant tissue parenchyma through lymphatics and hematogenous routes, often resulting in the manifestation of overt, clinically detectable metastatic lesions. The occurrence of poorly differentiated carcinoma metastasizing to the nervous system is rare, with a limited number of cases discussed in the literature [6,7,8]. In our literature search for cases of metastatic poorly differentiated epithelioid carcinoma across various age groups, we encountered a comprehensive report by Pentheroudakis et al. [9]. This report compiled data from 24 retrospective studies, including a cohort of 689 patients, with data spanning from 1975 to 2006. These studies show that poorly differentiated carcinoma of unknown primary (CUPAx) exhibited a predilection for women, with an average age of 52 years.

Poorly differentiated epithelial carcinoma is characterized histologically as epithelioid cells (round to oval cells) with a nesting arrangement and a desmoplastic stroma with feeding vessels separating pleomorphic to anaplastic appearing tumor cell nests. Microscopically, our cases’ specimen was a cellular, cytologically malignant spindle cell neoplasm composed predominantly of epithelioid cells with large atypical vesicular nuclei and prominent nucleoli with somewhat amphophilic cytoplasm.

The broad category of neoplasia, such as carcinoma, sarcoma, lymphoma, or melanoma is determined by immunohistochemical staining. Employing a screening panel to assess the expression of markers representing major lineages (including epithelial, mesenchymal, lymphoid, and melanocytic) often serves as the diagnostic indicator [2]. In a study involving 98 poorly differentiated epithelial tumors, the utilization of AE1, CAM 5.2, and EMA markers in pairs significantly improved the recognition of epithelial differentiation, achieving a sensitivity rate ranging from 80% to 99% [1]. This highlights the heterogeneity in biomarker expression among poorly differentiated carcinomas and the importance of employing all three biomarkers to enhance screening sensitivity. In these cases, staining with CK (cytokeratin) was characterized by a diffuse and robust pattern, particularly in carcinomas or sarcomatoid carcinomas [2].

In another notable study involving 87 patients, where initial diagnoses of poorly differentiated carcinoma (PDC) or adenocarcinoma (PDA) were established through light microscopic examination, the pivotal role of immunoperoxidase staining confirmed the diagnosis [3]. This staining method successfully confirmed the presence of poorly differentiated carcinoma in 49 patients (56%). It also unveiled an alternative diagnosis in 14 patients (16%), including melanoma in eight cases, lymphoma in four, prostatic carcinoma in one, and yolk sac carcinoma in one. These findings underscore immunoperoxidase staining imperativeness to rule out dedifferentiation in various lineages, which can closely mimic the histological appearance of PDC.

Our cases’ specimen initially showed multifocal strong positivity for Sry-related HMg-Box gene 10 (SOX10), pan-keratin, and CAM5.2 along with negativity for S100, HMB45, and Mart-1. Multifocal positivity was also found in claudin-4 with a distinctively membranous pattern, strongly suggesting epithelial differentiation.

Nuclear transcription factor SOX10 plays an important role in melanocytic cell differentiation and has been shown to be a sensitive marker of spindle and desmoplastic melanoma. It is useful for identifying metastatic melanoma in sentinel lymph nodes and higher expression of SOX10 has been associated with melanoma progression and aggressiveness [10].

In our literature search for cases of dedifferentiated metastatic melanoma across various age groups, we came across a report by Agaimy et al., which included 35 cases of dedifferentiated and undifferentiated melanomas, along with a review of 50 previously reported cases [11]. In seven of these cases, the initial diagnosis was an unclassified epithelioid malignancy, and in two cases, it was poorly differentiated carcinoma. After immunohistochemical analysis, these diagnoses were reclassified to be dedifferentiated melanoma. Within this group, the axillary lymph node emerged as a significant metastatic site, along with involvement of the lung soft tissue. This tumor also showed a multifocal spreading pattern, often involving the axilla, inguinal region, neck, and digestive system. 

In our specific case, the significant heterogeneity observed in biomarker expression, along with the diffuse positivity for SOX10, the axillary location of the lesion, multifocal spread to the nervous system with associated incontinence, and the presence of B-symptoms, collectively contributed to a broad and challenging differential diagnosis. Consequently, this complexity in diagnosis extended the patient’s hospital stay.

Notably, the multifocal expression of CAM5.2 and claudin-4, with a predominance of epithelioid cells, lent support to the hypothesis of an epithelial lineage, ultimately leading to the classification as metastatic poorly differentiated epithelial carcinoma. However, it is important to exercise caution as dedifferentiated melanoma cannot be definitively ruled out in this context. Due to the intricate nature of poorly differentiated tumors, thorough evaluation and consideration of all possible diagnostic lineages will aid in appropriate management. Further immunohistochemical study (ALK, MITF, myeloperoxidase, and HepPar1) may have helped establish diagnosis in this case.

Genomic analysis was explored to help distinguish between carcinoma and other lineages such as melanoma, lymphoma, and sarcoma. Our case showed MAP2K1 gene mutations and p16INK4a loss, which are commonly found in melanoma malignancies [12,13]. MYC amplification, seen in our case, is common in solid tumors including breast and liver adenocarcinoma; however, it is also prevalent in cutaneous melanoma [14]. CDKN2A/B CDKN2B loss has been associated with high-grade salivary gland carcinomas and was also positive in our case [15]. These findings exemplify the overlap between carcinoma and melanoma and underscore the diagnostic challenges in differentiating this high-grade tumor.

Distinguishing poorly differentiated carcinoma from dedifferentiated tumors, such as lymphoma, melanoma, and mesenchymal tumors, can be challenging. Diagnostic hurdles include overlapping features and shared morphologic and immunophenotypic characteristics. For instance, melanoma can exhibit epithelioid features at the microstructural level, and while cytokeratin positivity may be observed, diffuse positivity is infrequent. Furthermore, metastatic melanoma has the potential to undergo complete dedifferentiation, resulting in lost expression of all melanocytic markers, frequently leading to misdiagnoses as poorly differentiated carcinoma [16]. Before the routine use of immunohistochemistry in clinical practice, many poorly differentiated carcinomas were treated with cisplatin. Studies, such as the one conducted by Hainsworth, have shown that few lesions initially diagnosed as poorly differentiated carcinoma and responded to cisplatin therapy were subsequently re-evaluated and confirmed to be cases of melanoma [3]. 

In summary, we describe a case of poorly differentiated epithelioid carcinoma in a patient with an axillary mass, fecal and urinary incontinence, leg weakness, and B-symptoms. Although this diagnosis is often associated with cancer of unknown primary, it is atypical to see poorly differentiated carcinoma metastasize to the axillary node. Staining with immunoperoxidase, identifying biomarker sensitivity, and utilizing genomic analysis can help narrow the differential diagnosis and ensure appropriate treatment initiation. High clinical suspicion and early biopsy in patients with an axillary mass, B-symptoms, and acute focal deficits is key to preventing a prolonged hospital course. Despite narrowing the differential, ruling out dedifferentiated melanoma and poorly differentiated carcinoma with diagnostic certainty may be unachievable without a primary melanoma source. We hope this case contributes to the limited literature available on poorly differentiated epithelial carcinoma versus dedifferentiated metastatic melanoma to expand our understanding of these aggressive entities.

## Figures and Tables

**Figure 1 diseases-12-00196-f001:**
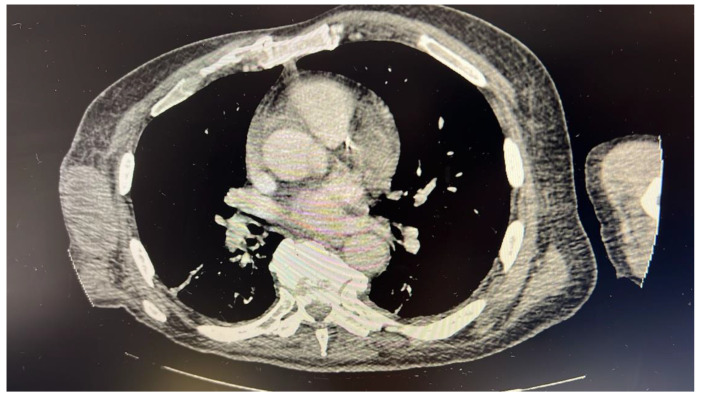
Computed tomography scan showing right axillary soft tissue mass measuring 8.3 × 3.9 × 6.5 cm with multiple prominent retroperitoneal lymph nodes.

**Figure 2 diseases-12-00196-f002:**
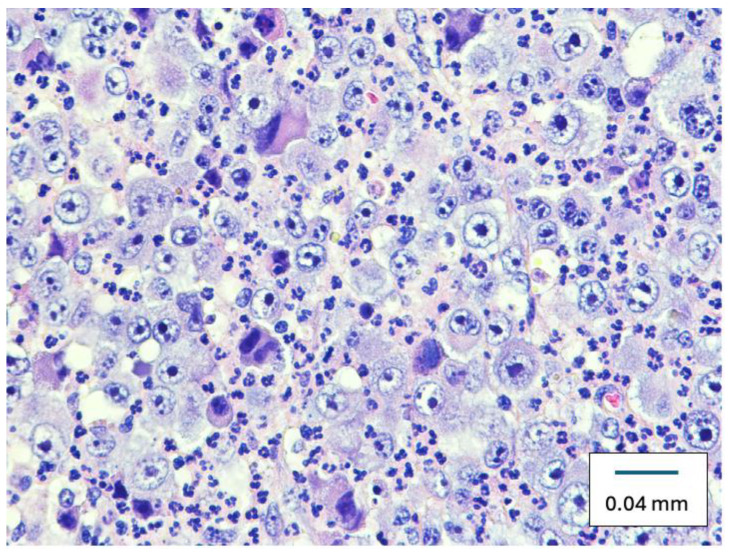
Right axillary mass specimen showing large, atypical spindle cells with large atypical vesicular nuclei, eosinophilic to amphophilic cytoplasm, and prominent nucleoli. (Original magnification ×400, hematoxylin and eosin stain).

**Figure 3 diseases-12-00196-f003:**
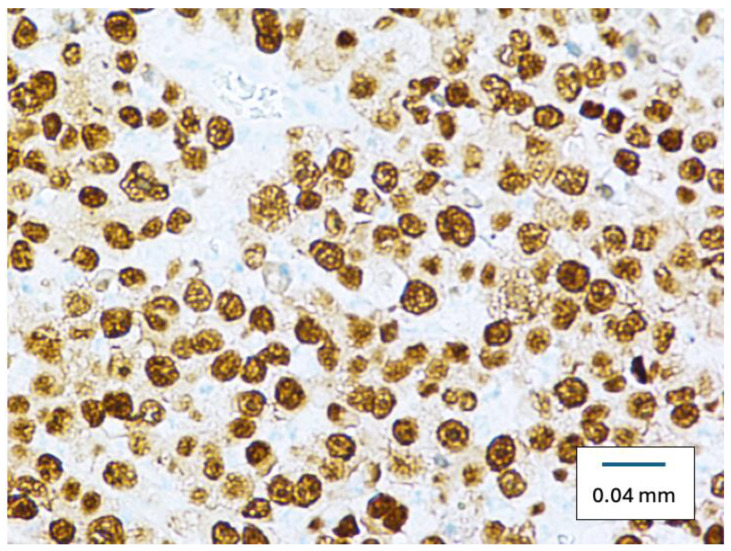
Right axillary mass specimen showing positive SOX 10 staining. (Original magnification ×400).

**Figure 4 diseases-12-00196-f004:**
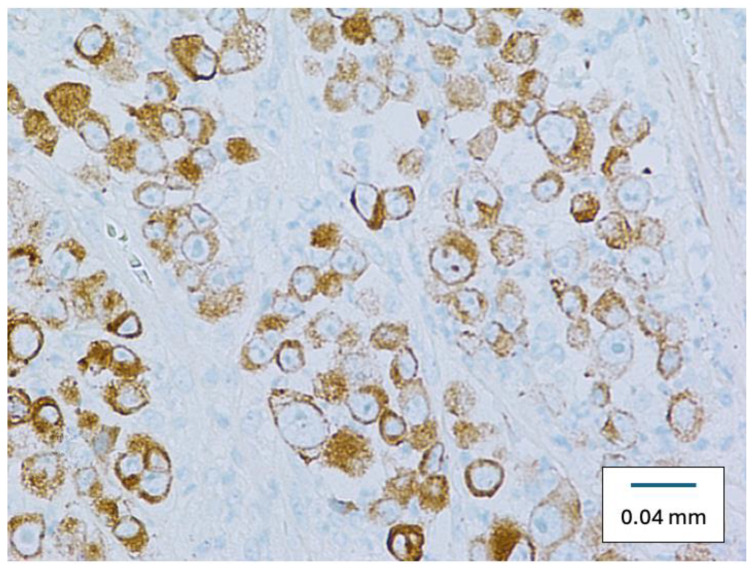
Right axillary mass specimen showing positive CAM-52 staining. (Original magnification ×400).

**Table 1 diseases-12-00196-t001:** Initial pathology report.

Positive Staining	Negative Staining
WT1 SOX10 CAM5.2 CD45 ^a^	AE1/AE3 CK7 CK20 TTF-1 Melan-A HMB45 CDX2 CD138 Chromogranin Synaptophysin S100 Mammaglobin GATA3 GCDFP

^a^ CD45 was positive on only a few lymphocytes but negative on the neoplastic cells themselves.

**Table 2 diseases-12-00196-t002:** Pathology report from an outside institution.

Positive Staining	Negative Staining
SOX10 Pan-keratin CAM5.2 Claudin-4 INI-1 SMARCA4 PRAME	S100 HMB45 Mart-1 NRASQ61R BerEp4 CK7 CD30 GFAP GATA3 BRAFv600E p63m EMA

**Table 3 diseases-12-00196-t003:** Genomic analysis from Foundation Medicine, Inc.

Positive Findings
MAP2K1 (MEK1) E102_I103del
MYC amplification
CDKN2A/B CDKN2B loss
p16INK4a loss and p14ARF loss in exons 2-3
PIK3C2G E1231K
RB1 Q395*
TERT promoter -146C>T
TET2 D1788fs*32
ATR
V1689M
ATRX amplification
FAM123B
R21H
KIT
T67S
PDGFRB rearrangement
ZNF703
A401_H402insPTHLGGSSCST CSA

## Data Availability

No datasets were used to report findings in this case.

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
