# Peer review of "Malignant Epithelioid Neoplasm versus Dedifferentiated Malignant Melanoma: A Case Report"

_diseases, 2024, doi:10.3390/diseases12090196_

Round 1

Reviewer 1 Report

Comments and Suggestions for Authors

This manuscript summarize a case of a 75-year-old male, who had a diagnosis of a poorly differentiated epithelial carcinoma. They describe that the case underscores the morphological and immunoperoxidase staining similarities between poorly differentiated carcinoma and de-differentiated tumors of varying origins. Their results highlight the intricate nature of these malignant metastatic tumors and their overlapping manifestations- which requires provider awareness. The case contributes to the limited available cases on poorly differ-entiated epithelial carcinoma versus dedifferentiated metastatic melanoma and definitely it will expand the understanding of this aggressive carcinoma.

Comment: Using references in the abstract is not usual.

I did not fint the ethical permission of the patient.

I assume that the authors received the already stained slide because the Materials and methods are missing. This should be mention ...

Author Response

Comments 1: 

Please add a scale bar.

Response 1: A scale bar was added to figures 1-3.

Comments 2:

In this section, you should add the Institutional Review Board Statement and approval number, if relevant to your study. You might choose to exclude this statement if the study did not require ethical approval. Please note that the Editorial Office might ask you for further information. Please add “The study was conducted in accordance with the Declaration of Helsinki, and approved by the Institutional Review Board (or Ethics Committee) of NAME OF INSTITUTE (protocol code XXX and date of approval).” for studies involving humans. OR “The animal study protocol was approved by the Institutional Review Board (or Ethics Committee) of NAME OF INSTITUTE (protocol code XXX and date of approval).” for studies involving animals. OR “Ethical review and approval were waived for this study due to REASON (please provide a detailed justification).” OR “Not applicable” for studies not involving humans or animals.

Response 2: IRB approval was not necessary in the case report, no experimental study was preformed. This is addressed in line 46-47.

Comments 3: 

Any research article describing a study involving humans should contain this statement. Please add “Informed consent was obtained from all subjects involved in the study.” OR “Patient consent was waived due to REASON (please provide a detailed justification).” OR “Not applicable.” for studies not involving humans. You might also choose to exclude this statement if the study did not involve humans.

Written informed consent for publication must be obtained from participating patients who can be identified (including by the patients themselves). Please state “Written informed consent has been obtained from the patient(s) to publish this paper” if applicable.

Response 3: A statement that informed consent was obtained by next of kin for deceased patient was added in line 48-49.

Comments 4:

We encourage all authors of articles published in MDPI journals to share their research data. In this section, please provide details regarding where data supporting reported results can be found, including links to publicly archived datasets analyzed or generated during the study. Where no new data were created, or where data is unavailable due to privacy or ethical restrictions, a statement is still required. Suggested Data Availability Statements are available in section “MDPI Research Data Policies” at https://www.mdpi.com/ethics.

Response 4: No datasets were used to report findings in this case, as stated in line 50.

Reviewer 2 Report

Comments and Suggestions for Authors

There are several comments.

1. Please check the result of pan-keratin (positive in text but negative in Table 1).

2. Please check cytologically malignant spindle cell neoplasm or malignant epithelioid cell neoplasm.

3. In this case, leukocytosis (43,000), thrombocythemia (841,000), anemia (hemoglobin 6.6), and thrombophilia (1000) were identified. Which conditions can these abnormal laboratory findings be found? 

4.  Further immunohistochemical study (for example, ALK, MITF, myeloperoxidase, TTF-1, HepPar1, etc.)  is recommended for accurate diagnosis.

5. It would be better to describe radiological findings.

Comments on the Quality of English Language

Please check English grammar.

Author Response

Response 2

  1. Pan-keratin was not included for analysis in the initial pathology report (Table I). We sent our specimen for subsequent review at an outside pathology department, whose report included positive pan-keratin staining (Table II).

  1. Is there more to this question? The specimen was described cytologically as a malignant spindle cell neoplasm and was classified as a malignant epithelioid cell neoplasm given the staining. (Lines 110-116).

  1. Myeloproliferative neoplasms are rare blood cancers that cause the body to overproduce blood cells. The three main subcategories are essential thrombocythemia, myelofibrosis, and polycythemia vera. Often these patients have anemia of chronic disease, with low hemoglobin. We added a statement describing this to lines 88-89.

  1. TTF-1 was included in the initial pathology report (Table 1), which was negative. The other immunohistochemical stains were not included in the pathology report and we addressed this by adding a statement to lines 229-230.

  1. Radiologic findings in a computed tomography scan showed a right axillary soft tissue mass measuring 8.3 x 3.9 x 6.5 cm with multiple prominent retroperitoneal lymph nodes. This was added to lines 89-92, and the figure of the axillary mass on CT scan was added as well. (new Figure I)

Reviewer 3 Report

Comments and Suggestions for Authors

Cancer is a very big problem nowadays. Mortality is constantly increasing, so quick and appropriate diagnostics are very important. Especially since some symptoms of the disease are not clear. Therefore, case report articles constitute an important scientific component.

The author describes the case and the diagnostic process in detail.

The work also contains a discussion part and a summary.

The only complaint is the bibliography which is not very extensive and not very new.

Despite this, the publication meets the requirements.

Author Response

Response 3

Thank you for your comment. Despite the case not addressing something very new, this topic is scarce in the literature albeit important for patients with this aggressive tumor. This case is beneficial to future research proposing guidelines for diagnosing and managing such enigmatic malignancies.

Round 2

Reviewer 2 Report

Comments and Suggestions for Authors

There are two minor comments.

1. The manuscript describes "Microscopically, the specimen was a cellular, cytologically malignant spindle cell neoplasm...". It would be better to include microscopic images of the malignant spindle cells in the Figure. This addition will help readers better understand the cytological characteristics of the lesion.

2. The manuscript includes genomic analysis results. It would be better explain the meaning and implications of these results.

Comments on the Quality of English Language

Please check English grammar.

Author Response

  1. The manuscript describes "Microscopically, the specimen was a cellular, cytologically malignant spindle cell neoplasm...". It would be better to include microscopic images of the malignant spindle cells in the Figure. This addition will help readers better understand the cytological characteristics of the lesion.

The specimen is shown in Figure II and the description was updated to improve readers understanding of the cytological characteristics of the lesion.

  1. The manuscript includes genomic analysis results. It would be better explain the meaning and implications of these results.

In cases of poorly differentiated neoplasm, it is difficult to distinguish between carcinoma and other cancers such as melanoma, lymphoma, and sarcoma. Genomic analysis may help identify the lineage in these cases. We included more details of genomic analysis results and meaning and implications of these results. Update can be found in lines 233 - 240.